# Aux-NAS: Exploiting Auxiliary Labels with Negligibly Extra Inference Cost

**Yuan Gao**[1], **Weizhong Zhang**[2], **Wenhan Luo**[3], **Lin Ma**[4], **Jin-Gang Yu**[5], **Gui-Song Xia**[1]*, **Jiayi Ma**[1]
[1]Wuhan University, [2]Fudan University, [3]HKUST, [4]Meituan, [5]South China University of Technology
`{ethan.y.gao, whluo.china, forest.linma, jyma2010}@gmail.com,`
`weizhongzhang@fudan.edu.cn, jingangyu@scut.edu.cn, guisong.xia@whu.edu.cn`

## Abstract

We aim at exploiting additional auxiliary labels from an independent (auxiliary) task to *boost the primary task performance* which we focus on, while preserving *a single task inference cost* of the primary task. While most existing auxiliary learning methods are optimization-based relying on loss weights/gradients manipulation, our method is architecture-based with a flexible *asymmetric structure* for the primary and auxiliary tasks, which produces different networks for training and inference. Specifically, starting from two single task networks/branches (each representing a task), we propose a novel method with evolving networks where only primary-to-auxiliary links exist as the cross-task connections after convergence. These connections can be removed during the primary task inference, resulting in a single-task inference cost. We achieve this by formulating a Neural Architecture Search (NAS) problem, where we initialize bi-directional connections in the search space and guide the NAS optimization converging to an architecture with only the single-side primary-to-auxiliary connections. Moreover, our method can be incorporated with optimization-based auxiliary learning approaches. Extensive experiments with *six* tasks on NYU v2, CityScapes, and Taskonomy datasets using VGG, ResNet, and ViT backbones validate the promising performance. The codes are available at https://github.com/ethanygao/Aux-NAS.

## 1 Introduction

In this paper, we tackle the practical issue of auxiliary learning, which involves improving the performance of a specific task (*i.e.*, the primary task) while incorporating additional auxiliary labels from different tasks (*i.e.*, the auxiliary tasks). We aim to efficiently leverage these auxiliary labels to enhance the primary task's performance while maintaining a comparable computational and parameter cost to a single-task network when evaluating the primary task.

Our problem is closely related to Multi-Task Learning (MTL) but has two distinct practical requirements: 1) only the performance of the primary task, rather than that of all the tasks, is of our interest, and 2) we aim to maintain a single-task cost during the inference. For example, we may primarily concerned with the semantic segmentation performance but have auxiliary depth labels available. The question is that *can we leverage those auxiliary labels to improve the primary task, while preserving the inference cost for the primary task similar to a single-task network*?

The primary obstacle to achieving these goals is the presence of the inherent *negative transfer*, which is caused by the conflict gradients from different tasks that flow to the shared layer Ruder (2017); Ruder et al. (2019); Vandenhende et al. (2020b). To alleviate the negative transfer, most existing auxiliary learning methods are optimization-based, which formulates a shared network and deals with the conflicted gradients to the shared layer by modifying the auxiliary loss weights or gradients Du et al. (2020); Liu et al. (2022); Navon et al. (2021); Verboven et al. (2020); Shi et al. (2020). However, two most recent studies independently show that it is challenging to solve the negative transfer solely by manipulating the loss weights or gradients Xin et al. (2022); Kurin et al. (2022).

---

*Corresponding Author.

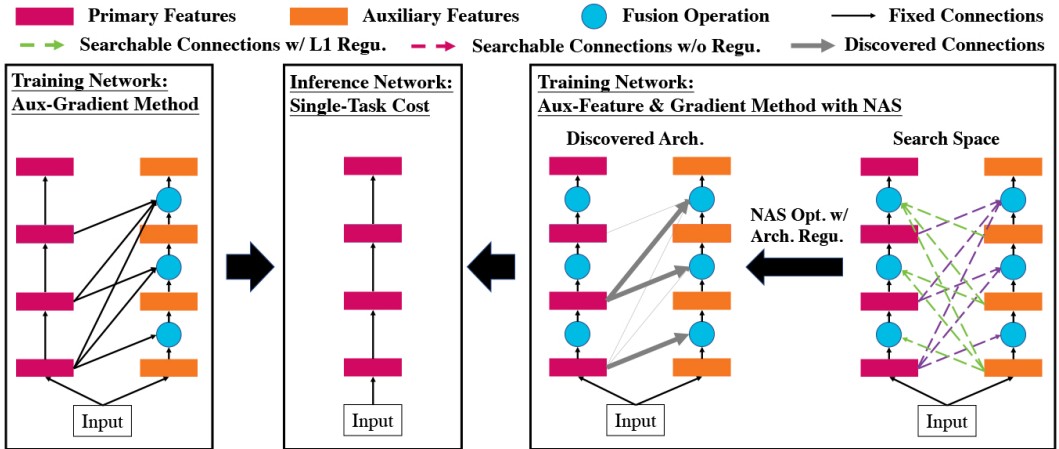

Figure 1: Overview of the proposed methods. Our methods are based on an asymmetric architecture that employs different networks for training and inference, where we exploit gradients and/or features from the auxiliary task during the training, and preserve a single-task cost for evaluating the primary task. Our first method (Leftmost) leverages the auxiliary gradients. Our second method (Rightmost) exploits both auxiliary features and gradients, where the auxiliary-to-primary connections (green dash lines) are gradually pruned out by NAS, resulting in a converged architecture with only primary-to-auxiliary connections (the line widths indicate the converged architecture weights). Finally, the primary-to-auxiliary connections, as well as the auxiliary branch, can be safely removed to obtain a single task network (Middle) to inference the primary task. The network arrows indicate the directions/inverse directions of the feature/gradient flow. (Best view in colors.)

Instead, the architecture-based methods with soft parameter sharing assign separated task-specific model parameters, which avoids the conflicting gradients from the root Shi et al. (2023). Therefore, we aim to better solve the negative transfer using the soft parameter sharing based architectures. In other words, we propose to learn a separate set of parameters/features for each task to avoid the negative transfer, where different primary and auxiliary features interact with each other. However, it is challenging to achieve a single-task inference cost with separate parameters for different tasks, as the primary and auxiliary networks rely on each other as input to generate higher-level features.

The above analysis inspires us to design *an asymmetric network architecture that produces changeable networks between the training and the inference phases*, *i.e.*, the training network can be more complex to better exploit the auxiliary labels, while those auxiliary-related computations can be removed during the inference. Starting with multiple single-task branches (each for one task), our key design is to ensure that *the network is asymmetric and only includes directed acyclic primary-to-auxiliary forward connections*. This allows us to safely remove inter-task connections during primary task inference, resulting in a multi-task level performance and a single-task inference cost.

Motivated by this, our first method exploits the **gradients** from the auxiliary task as extra regularization to the primary task, while those auxiliary computations can be removed during the inference as the gradients are no longer required. We implement this by establishing multiple layerwise forward connections from the primary to the auxiliary tasks, where the auxiliary task leverages the features from, and thus back-propagates the gradients to, the primary task, as shown in Fig. 1 (Left).

The follow-up question is that can we harness **both the features and gradients** from the auxiliary task during the training, while still maintaining a single-task cost in the inference? Fortunately, this can be achieved by training an evolving network with a novel Neural Architecture Search (NAS) algorithm that employs asymmetric constraints for different architecture weights. Specifically, the NAS search space is initialized to include all bi-directional primary-to-auxiliary and auxiliary-to-primary connections. We then impose $\ell_1$ constrains on the auxiliary-to-primary architecture weights to gradually prune them out during the search procedure. We illustrate this in Fig. 1 (Right).

Both of our proposed methods are general-applicable to **various primary-auxiliary task combinations**[1] with **different single task backbones**. Moreover, the proposed methods **are orthogonal to, and can be incorporated with, most existing optimization-based auxiliary learning methods**

---

[1] Specifically, given the primary and auxiliary task(s), we do not assume how much they are related. Instead, our methods are designed to automatically learn what to share between the tasks.

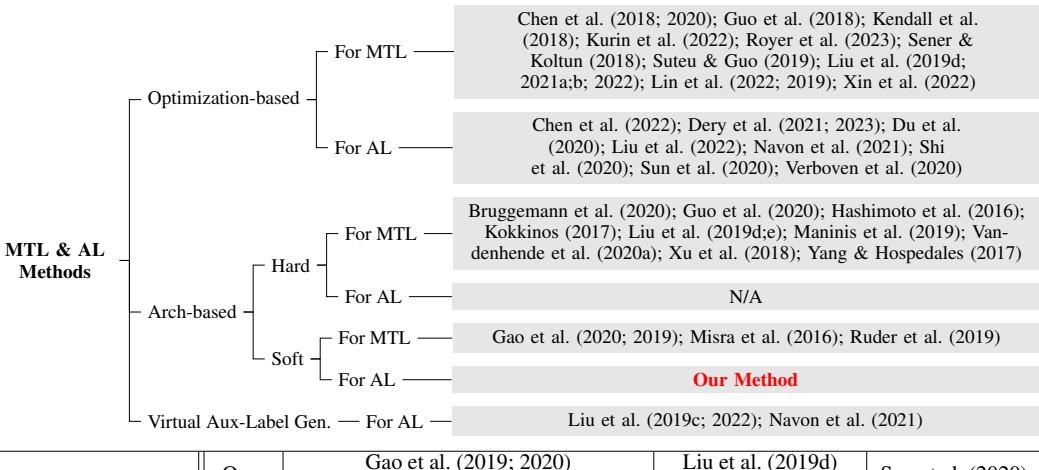

| Arch-based Methods | Ours | Gao et al. (2019; 2020) Misra et al. (2016); Ruder et al. (2019) | Liu et al. (2019d) Maninis et al. (2019) | Sun et al. (2020) |
|---|---|---|---|---|
| | (Soft) | (Soft) | (Hard) | (Hard) |
| Inference FLOPs | $N$ | $(K+1)N + (K+1)KM/2$ | $N+M$ | $\leq N$ |

Table 1: The taxonomy of our method in the MTL and AL areas (Top), and the inference FLOPs of our method and the representative architecture-based ones (Bottom). AL, Soft/Hard mean Auxiliary Learning, Soft/Hard Parameter Sharing. $N$ is the FLOPs for a single task, $K$ is the number of auxiliary tasks, and $M$ is the fusion FLOPs for each task pair in Gao et al. (2019; 2020); Misra et al. (2016); Ruder et al. (2019), or the extra attention FLOPs in Liu et al. (2019d); Maninis et al. (2019).

Du et al. (2020); Liu et al. (2022); Navon et al. (2021); Verboven et al. (2020); Shi et al. (2020). We validate our methods with 6 tasks (see Sect. 4), using VGG-16, ResNet-50, and ViTBase backbones, on the NYU v2, CityScapes, and Taskonomy datasets. Our contributions are three-fold:

- We tackle the auxiliary learning problem with a novel asymmetric architecture, which produces different networks for training and inference, facilitating a multi-task level performance with a single-task level computations/parameters.

- We implement the above idea with a novel training architecture with layerwise primary-to-auxiliary forward connections, where the auxiliary task computations provide additional gradients and can be safely removed during the inference.

- We propose a more advanced method with evolving architectures to leverage both the auxiliary features and gradients, where the auxiliary-to-primary connections can be gradually cut off by a NAS algorithm with a novel search space and asymmetric regularizations.

## 1.1 TAXONOMY OF OUR METHODS

The taxonomy of our methods in both MTL and auxiliary learning areas is illustrated in Table 1. Our methods fall under the category of *architecture-based methods* for *auxiliary learning* with a *soft-parameter sharing* scheme. We note that the virtual aux-label generation methods operate in a distinct context from ours without auxiliary labels, and thus are beyond our scope.

In contrast to all the existing auxiliary learning methods which focus on designing effective optimization strategies, we instead to explore novel auxiliary learning architecture design, for which its objective can be freely integrated with any optimization-based methods, as validated in Sect. 4.1.

While architecture-based methods are generally better at migrating negative transfer Shi et al. (2023), there is few (if any) approach designed for auxiliary learning except for ours . Moreover, our methods leverage soft parameter sharing, which further alleviates the negative transfer compared with its hard parameter sharing counterpart Ruder (2017), by implementing independent model parameters. While independent model parameters often result in an increased inference cost as outlined in Table 1 (Bottom), our methods are different from them by preserving a single-task inference cost.

## 2 RELATED WORK

**Multi-Task Learning.** MTL aims to improve the performance of all input tasks Long et al. (2017); Kokkinos (2017); Zamir et al. (2018), which can be categorized into Multi-Task Optimizations

(MTO) and Multi-Task Architectures (MTA) Ruder (2017); Vandenhende et al. (2020b). The multi-task optimization methods manipulate the task gradients/loss weights to tackle the negative transfer Kendall et al. (2018); Chen et al. (2018); Liu et al. (2019d); Lin et al. (2022); Chen et al. (2020); Guo et al. (2018); Sener & Koltun (2018); Lin et al. (2019); Liu et al. (2021b); Suteu & Guo (2019); Liu et al. (2021a; 2022); Royer et al. (2023). Our methods are orthogonal to MTO methods and follow the MTA category, *i.e.*, learning better features for different tasks via elaborated Hard or Soft Parameter Sharing (HPS or SPS) network architectures Misra et al. (2016); Ruder (2017); Xu et al. (2018); Ruder et al. (2019); Gao et al. (2019); Maninis et al. (2019); Gao et al. (2020); Liu et al. (2019d); Vandenhende et al. (2020b;a); Guo et al. (2020); Bruggemann et al. (2020); Yang & Hospedales (2017); Kokkinos (2017); Hashimoto et al. (2016); Liu et al. (2019e). Our methods leverage SPS scheme Misra et al. (2016); Ruder et al. (2019); Gao et al. (2019; 2020), which uses separate model weights for each task to better tackle the negative transfer with a single-task inference cost.

**Auxiliary Learning.** Most existing auxiliary learning methods are optimization-based, which use a shared feature set with auxiliary gradients/loss weights manipulation Liebel & Körner (2018); Du et al. (2020); Navon et al. (2021); Verboven et al. (2020); Shi et al. (2020); Liu et al. (2022); Chen et al. (2022); Sun et al. (2020); Dery et al. (2021). Most recently, Dery et al. (2023) proposed to search for appropriate auxiliary objectives from a candidate pool and achieves remarkable performance. Instead, our methods learn a unique feature set for each task, which can integrate with those methods. Recent works generate virtual auxiliary labels Liu et al. (2019c); Navon et al. (2021); Liu et al. (2022), which have a very different setting from ours and is thus beyond our scope.

**Network Pruning & Neural Architecture Search.** Network pruning aims at removing unimportant layers without severely deteriorating the performance Han et al. (2016); He et al. (2017); Liu et al. (2017); Luo et al. (2017); Ye et al. (2018); Gordon et al. (2018). As the pruning process is crucial for the final performance Frankle & Carbin (2019), our algorithm gradually prunes primary-to-auxiliary connections by a single-shot gradient based NAS method Guo et al. (2019); Pham et al. (2018); Saxena & Verbeek (2016); Bender et al. (2018); Liu et al. (2019b); Xie et al. (2019); Akimoto et al. (2019); Wu et al. (2019); Zhang et al. (2019); Liu et al. (2019a); Mei et al. (2020); Gao et al. (2020).

## 3 Methods

In this section, we propose two novel methods exploiting the auxiliary labels to enhance our primary task, while keeping a single task inference cost. Based on the soft parameter sharing architecture, our key design is *an asymmetric architecture* which creates different networks for training and inference, *i.e.*, the training architecture is more complex to exploit the auxiliary labels, while those auxiliary-related computations/parameters can be safely removed during the primary task inference.

In the following, we first discuss our asymmetric architecture design. Then, we implement two novel algorithms, where the first method exploits the auxiliary gradients, and the second method leverages both the auxiliary features and gradients in an evolving network trained by a novel NAS algorithm. After that, we implement the feature fusion operations. Finally, we provide a taxonomy of our methods within the areas of both MTL and auxiliary learning.

### 3.1 The Asymmetric Architecture with Soft Parameter Sharing

We tackle auxiliary learning by the architecture-based methods. Due to its merit of better migrating the negative transfer, our methods follow *the soft parameter sharing architecture*, where different tasks exhibit separated network branches (*i.e.*, independent/unshared network parameters) with feature fusion connections across them Ruder (2017); Vandenhende et al. (2020b).

Given a primary task and an auxiliary task, the widely used soft parameter sharing structure in MTL is shown in Fig. 2 (Left). Let $P_i^{\text{fea}}$ and $A_i^{\text{fea}}$ be the primary and auxiliary features for

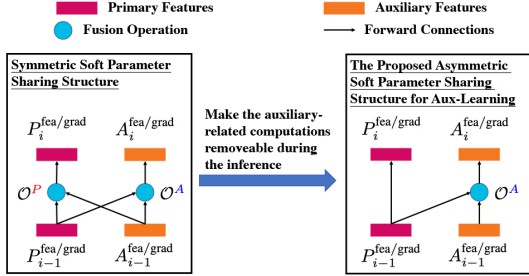

Figure 2: The asymmetric primary-auxiliary architecture with soft parameter sharing. (Best view in colors.)

the $i$-th layer, and $P_i^{\text{grad}}$ and $A_i^{\text{grad}}$ be the corresponding gradients, then the forward and the backward for Fig. 2 (Left) are Eqs. 1 - 4, where $\mathcal{O}^P = [\mathcal{O}^{PP}, \mathcal{O}^{PA}]$ and $\mathcal{O}^A = [\mathcal{O}^{AP}, \mathcal{O}^{AA}]$ are the learnable fusion operations parameterized by the model weights $\theta$, and $d\mathcal{O}^\cdot/d\theta$ are the corresponding derivatives. Equation 1 shows that the primary feature from the higher layer $i$ takes both the primary and the auxiliary features from the $i - 1$th layer. Thus, the auxiliary branch cannot be removed when inferencing the primary task in this case.

Symmetric, Fig. 2 (Left):

$$P_i^{\text{fea}} = \mathcal{O}^P[P_{i-1}^{\text{fea}}, A_{i-1}^{\text{fea}}]^\top = \mathcal{O}^{PP}P_{i-1}^{\text{fea}} + \mathcal{O}^{PA}A_{i-1}^{\text{fea}}, \quad (1)$$

$$A_i^{\text{fea}} = \mathcal{O}^A[P_{i-1}^{\text{fea}}, A_{i-1}^{\text{fea}}]^\top = \mathcal{O}^{AP}P_{i-1}^{\text{fea}} + \mathcal{O}^{AA}A_{i-1}^{\text{fea}}, \quad (2)$$

$$P_{i-1}^{\text{grad}} = \frac{d\mathcal{O}^{PP}}{d\theta}P_i^{\text{grad}} + \frac{d\mathcal{O}^{AP}}{d\theta}A_i^{\text{grad}}, \quad (3)$$

$$A_{i-1}^{\text{grad}} = \frac{d\mathcal{O}^{PA}}{d\theta}P_i^{\text{grad}} + \frac{d\mathcal{O}^{AA}}{d\theta}A_i^{\text{grad}}, \quad (4)$$

Asymmetric, Fig. 2 (Right) (Ours):

$$P_i^{\text{fea}} = P_{i-1}^{\text{fea}}, \quad (5)$$

$$A_i^{\text{fea}} = \mathcal{O}^A[P_{i-1}^{\text{fea}}, A_{i-1}^{\text{fea}}]^\top = \mathcal{O}^{AP}P_{i-1}^{\text{fea}} + \mathcal{O}^{AA}A_{i-1}^{\text{fea}}, \quad (6)$$

$$P_{i-1}^{\text{grad}} = P_i^{\text{grad}} + \frac{d\mathcal{O}^{AP}}{d\theta}A_i^{\text{grad}}, \quad (7)$$

$$A_{i-1}^{\text{grad}} = \frac{d\mathcal{O}^{AA}}{d\theta}A_i^{\text{grad}}. \quad (8)$$

Since the gradients are no longer required during the inference, we design an asymmetric soft parameter sharing structure, which enables removing the auxiliary computations during the inference. As shown in Fig. 2 (Right), we propose to only exploit the auxiliary gradients (rather than features) as additional regularization for the primary task. According to the corresponding forward (Eqs. 5 and 6) and backward (Eqs. 7 and 8), Eq. 7 shows the auxiliary gradients are used to train the primary task while Eq. 5 enables to maintain a single-task cost during the primary task inference.

**Remark 1** *The above analysis indicates that the structure for our problem should follow Fig. 2 (Right). We implement two methods in Sects. 3.2 and 3.3. Sect. 3.2 directly applies Fig. 2 (Right), while Sect. 3.3 establishes an evolving architecture that is initialized with bi-directional inter-task connections as Fig. 2 (Left), then the auxiliary-to-primary connections are gradually cut off using NAS during training, resulting in a converged structure as Fig. 2 (Right).*

## 3.2 THE AUXILIARY GRADIENT METHOD

Starting from two independent/unshared single task networks like Misra et al. (2016); Gao et al. (2019); Ruder et al. (2019), our first method *implement Eqs. 5 - 8 with moderate extension* by inserting **multiple layerwise** primary-to-auxiliary connections (representing the forward feature fusion) between the two branches. We denote this method as the Auxiliary Gradient method (Aux-G).

Our training architecture is shown in Fig. 1 (Left). We use a fusion operation on each layer of the auxiliary branch, which takes multiple features as input and produces a single output feature. As a result, it enables multiple inter-task connections pointing to the same sink node of the auxiliary branch, which in turn allows multiple gradients routing to the primary branch.

Denoting the auxiliary task feature from the $(i - 1)$-th layer as $A_{i-1}$, and the primary feature from the $j$-th layer as $P_j$ where $j \leq i$, the fused auxiliary feature at $i$-th layer $A_i$ is:

$$A_i = \mathcal{O}^A\Big(A_{i-1}, \alpha_{0,i}P_{0,i}, ..., \alpha_{i-1,i}P_{i-1,i}\Big), \quad (9)$$

where $\mathcal{O}$ is the fusion operation, whose implementation will be discussed in Sect. 3.4. $\alpha_{j,i}$ is a binary indicator representing the location of the primary-to-auxiliary connection, *i.e.*, $\alpha_{j,i}$ is 1 if there exists a connection from $j$-th primary layer to $i$-th auxiliary layer, otherwise, $\alpha_{j,i}$ is 0.

## 3.3 THE AUXILIARY FEATURE AND GRADIENT METHOD WITH NAS

We further extend Aux-G to *exploit both auxiliary features and gradients via an evolving architecture trained by a novel NAS algorithm.* Being initialized with bi-directional connections like Fig. 2 (Left), our NAS method guides the architecture converging to Fig. 2 (Right) with only primary-to-auxiliary connections, which can be removed during the primary task inference. We denote this method as the Auxiliary Feature and Gradient method with NAS (Aux-NAS).

**Remark 2** *Our search space applies **bi-directional** connections (primary-to-auxiliary and auxiliary-to-primary) at every layer between the two fixed single task backbones.*

Such search space design enables us to exploit both the features and the gradients from the auxiliary task, which also exhibits two additional merits: i) the proposed search space is general-applicable to

any primary and auxiliary task combinations as it searches the general feature fusing scheme, and ii) it efficiently exploits the layerwise features of each task without introducing negative transfer.

**Remark 3** *We also propose a novel search algorithm that facilitates the evolving architectures converging to a model where only **the primary-to-auxiliary connections** exist between the tasks. Therefore, those connections, as well as the auxiliary branch, can be safely removed without affecting the primary task performance during the inference.*

By such a design, we implicitly assume that besides the discovered discrete architecture, it is also important to exploit the mixed models (where the architecture weights are between 0 and 1) in the training procedure during the search phase. We note that, however, such importance of leveraging the mixed-model training was not fully exploited in the popular one-shot gradient based NAS algorithms (*e.g.*DARTS Liu et al. (2019b)). Specifically, the most widely used training procedure of a typical one-shot gradient based NAS algorithm includes a search phase and a retrain phase[2], which produces the inconsistent learning objectives between those two phases. As a consequence, performance gap or generalization issue is witnessed between the discovered mixed architecture and the retrained single architecture Xie et al. (2019); Li et al. (2019).

Our method does not suffer from this issue, because all the primary-to-auxiliary connections are cut off in our evaluation, regardless of a mixture model or a single model they converge to. The only requirement of our method is to restrict the architecture weights associated with the auxiliary-to-primary connections converging to a small value. Our NAS algorithm without a retrain phase makes it meaningful to exploit features and gradients from the auxiliary branch during the search phase, as the searched model (with the auxiliary-related computations removed) is directly used for evaluation. We achieve this using a $\ell_1$ *regularization on the auxiliary-to-primary architecture weights*, which gradually cuts off all the auxiliary-to-primary during the search phase. We do not impose constraints on the primary-to-auxiliary connections. This method is shown in Fig. 1 (Right).

Formally, let $\boldsymbol{w}$ be the model weights, denote the architecture weights for the *auxiliary-to-primary* connections as $\boldsymbol{\alpha^P} = \{\alpha_{ij}^P, \forall(i,j) \text{ with } i \leq j\}$ where $\alpha_{ij}^P$ is for the connection from the *auxiliary* layer $i$ to the *primary* layer $j$, denote similarly the architecture weights for the *primary-to-auxiliary* connections as $\boldsymbol{\alpha^A} = \{\alpha_{ij}^A, \forall(i,j) \text{ with } i \leq j\}$, our optimization problem becomes:

$$\min_{\boldsymbol{\alpha^P}, \boldsymbol{\alpha^A}, \boldsymbol{w}} \mathcal{L}^{\mathcal{P}}(\mathbf{P}(\boldsymbol{\alpha^P}, \boldsymbol{w})) + \mathcal{L}^{\mathcal{A}}(\mathbf{A}(\boldsymbol{\alpha^A}, \boldsymbol{w})) + \mathcal{R}(\boldsymbol{\alpha^P}), \quad \text{with} \quad \mathcal{R}(\boldsymbol{\alpha^P}) = \lambda \|\boldsymbol{\alpha^P}\|_1, \quad (10)$$

where $\mathcal{L}^{\mathcal{P}}$ is the loss function for the primary task, $\mathcal{L}^{\mathcal{A}}$ is the loss function for the auxiliary task, $\mathcal{R}$ is the regularization term on $\boldsymbol{\alpha^P}$ with $\lambda$ as the regularization weight. $\mathbf{P} = \{P_i, \forall i\}$ are all the fused *primary* features with $P_i$ as that of the $i$-th layer, and $\mathbf{A} = \{A_i, \forall i\}$ are all the fused *auxiliary* features with $A_i$ as that of the $i$-th layer. Similar to Eq. 9, $P_i$ and $A_i$ are:

$$P_i(\boldsymbol{\alpha^P}, \boldsymbol{w}) = \mathcal{O}^P\Big(P_{i-1}, \alpha_{0,i}^P A_0, ..., \alpha_{i-1,i}^P A_{i-1}\Big), \quad (11)$$

$$A_i(\boldsymbol{\alpha^A}, \boldsymbol{w}) = \mathcal{O}^A\Big(A_{i-1}, \alpha_{0,i}^A P_0, ..., \alpha_{i-1,i}^A P_{i-1}\Big). \quad (12)$$

Note that, being different from Aux-G, NAS is necessary for this method to exploit the auxiliary features, as it is used to gradually cut off the auxiliary-to-primary connections.

## 3.4 FUSION OPERATION

We design a unique fusion operation for both of our methods, so as to better illustrate the merit of the proposed novel network connections. There are two principles to design the fusion operation: i) the fusion operation can take an arbitrary number of input features, and ii) there is a negligible cost on both parameters and computations in $P_i(\boldsymbol{\alpha^P}, \boldsymbol{w})$ of Eq. 11 with $\boldsymbol{\alpha^P}$ as all 0.

Existing researches discussed the fusion operations on several input features, such as those based on weighted-sum Misra et al. (2016), attention Liu et al. (2019d), and neural discriminative dimensionality reduction (NDDR) Gao et al. (2019), where both Liu et al. (2019d) and Gao et al. (2019) leverage 1x1 convolution for feature transformation. Besides the extension of taking arbitrary input features, our method also integrates the advantages of those methods, where we implement heavier

---

[2]The two-phase training procedure is needed because the search phase usually converges to a mixed model with the architecture weights between 0 and 1, thus the retrain phase comes up to prune the mixed model and retrain a single model with a fixed architecture for evaluation.

computations (*i.e.*, 1x1 convolution) on the inference-removable features and negligible computations (*e.g.*, BatchNorm and ReLU) on the feature that remains during the inference[3].

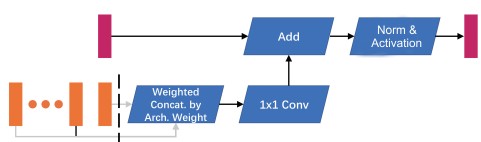

As show in Fig. 3, we implement the fusion operation in Eqs. 11 and 12 with *feature concatenation, 1x1 convolution, summation, normalization, and activation*, where [·] demotes a feature concatenation along the channel dimension, `Activ` and `Norm` can be ReLU and BatchNorm. Additionally, `BilinearInterp` is applied to each input feature when necessary before concatenation, it resizes the input features to the output spatial resolution, we omit it from Eqs. 13 and 14 for simplicity.

Figure 3: The illustration of the proposed fusion operator. Note that the auxiliary features (in orange color) are concatenated by its architecture weights. The dash line indicates that the regularized NAS objective in Eq. 10 enables to cut off the whole auxiliary computations (also the following 1x1 conv due to 0 input). (Best view in colors.)

$$P_i(\boldsymbol{\alpha^P}, \boldsymbol{w}) = \texttt{Activ}\Big(\texttt{Norm}\big(P_{i-1} + \texttt{1x1\_conv}([\alpha_{0,i}^P A_0, ..., \alpha_{i-1,i}^P A_{i-1}])\big)\Big) \quad (13)$$

$$A_i(\boldsymbol{\alpha^A}, \boldsymbol{w}) = \texttt{Activ}\Big(\texttt{Norm}\big(A_{i-1} + \texttt{1x1\_conv}([\alpha_{0,i}^A P_0, ..., \alpha_{i-1,i}^A P_{i-1}])\big)\Big) \quad (14)$$

Equation 13 enables to discard the heavier 1x1 convolution when $\boldsymbol{\alpha^P}$ are all 0 (as constrained by $\mathcal{R}(\boldsymbol{\alpha^P})$ in Eq. 10), which introduces negligible computation from a BatchNorm and a ReLU. Specifically, when introducing fusion operations in $n$ layers, there are only additionally $2n$ parameters (*i.e.*, $\beta$, $\gamma$ from BatchNorm), $2n$ summations and $2n$ productions (both from BatchNorm), and $n$ truncations (from ReLU). Those cost is negligible because at most $n$ is the number of layers in a network (*e.g.*, at most $n$ is 16 for VGG-16, 50 for ResNet-50, and 12 for ViTBase). Equation 14 does not introduce additional inference cost as it can be removed as a whole during the inference.

## 4 EXPERIMENTS

We fully assess our methods following the taxonomy in Table 1. We first show that our methods can be incorporated with the optimization-based methods. Then, we evaluate our methods against the architecture-based methods across various datasets, network backbones, and task combinations:

**Network Backbones.** We evaluate our methods on both CNNs and Transformers, *i.e.*, **VGG-16** Simonyan & Zisserman (2015), **ResNet-50** He et al. (2016), and **ViTBase** Dosovitskiy et al. (2021).

**Datasets.** We perform our experiments on the **NYU v2** Silberman et al. (2012), the **CityScapes** Cordts et al. (2016), and the **Taskonomy** Zamir et al. (2018) datasets.

**Tasks.** Six tasks are performed including **semantic segmentation, surface normal prediction, depth estimation, monocular disparity estimation, object classification**, and **scene classification**. We detail the task losses and the evaluation metrics in Appendix B.

We implement **Aux-G** with two granularities: **Aux-G-Stage** with linked connections at the last layer of each stage, and **Aux-G-Layer** with linked connections at every layer within each stage. We allow the bi-directional links for **Aux-NAS**. We restrict the fusion operation to only receive the features within 3 layers, *i.e.*, the $i$-th layer fusion operation only receive features of $i - 3$, $i - 2$, $i - 1$ layers from the other task, where $(\boldsymbol{\alpha^P}, \boldsymbol{\alpha^A})$ and $\boldsymbol{w}$ are trained alternatively by sampling two non-overlapped batches Gao et al. (2020). We give more details in Appendix C and analyze the architecture convergence in Appendix C.

### 4.1 COMPARISON WITH OPTIMIZATION-BASED METHODS

Our architecture-based methods are mathematically orthogonal to, and can be incorporated with, the optimization-based methods. In this section, we compare and incorporate our methods with four MTL methods: **Uncertainty** Kendall et al. (2018), **DWA** Liu et al. (2019d), **PCGrads** Yu et al. (2020), **CAGrads** Liu et al. (2021a), and one existing auxiliary learning **GCS** Du et al. (2020). *We also implement an auxiliary learning variant of* **PCGrads**, *i.e.*, **PCGrads-Aux**, *which always*

---

[3]Note that it is direct to include multiple candidate fusion operations in the search space using NAS, but chasing the fusion-op of each layer is problem-depend (thus distracting) and beyond the scope of this paper.

| NYU v2 | Primary: Seg | | Primary: Normal | | | | |
|---|---|---|---|---|---|---|---|
| | (%) (↑) | | Err (↓) | | | Within $t°$ (%) (↑) | |
| | mIoU | PAcc | Mean | Med. | RMSE | 11.25 | 22.5 |
| Single | 33.5 | 64.1 | 15.6 | 12.3 | 19.8 | 46.4 | 75.5 |
| Aux-Head | 34.7 | 65.4 | 15.3 | 11.7 | 20.0 | 48.4 | 75.9 |
| Adashare | 35.0 | 65.6 | 14.8 | 11.5 | 18.8 | 50.3 | 77.2 |
| Adashare-Aux | 35.0 | 65.7 | 14.4 | 11.1 | 19.1 | 51.2 | 78.3 |
| Aux-G-Stage | 35.4 | 65.9 | **14.1** | 10.7 | **18.4** | 52.0 | **79.0** |
| Aux-G-Layer | 35.6 | 65.9 | 14.4 | 11.0 | 18.8 | 50.8 | 78.7 |
| Aux-NAS | **36.0** | **66.1** | 14.2 | **10.6** | 18.9 | **52.4** | **79.0** |

Table 3: **Semantic segmentation** and **Surface normal prediction** on NYU v2 using **VGG-16**, with the other task as the auxiliary task.

| CityScapes | (%) (↑) | |
|---|---|---|
| Primary: Seg | mIoU | PAcc |
| Single | 68.3 | 94.5 |
| Aux-Head | 70.0 | 94.6 |
| Adashare | 70.3 | 94.7 |
| Adashare-Aux | 70.1 | 94.8 |
| Aux-G-Stage | 70.1 | 94.8 |
| Aux-G-Layer | 70.2 | 94.8 |
| Aux-NAS | **71.1** | **95.0** |

Table 4: Semantic seg. on **CityScapes** using **VGG-16** with disparity as aux. task.

*projects the auxiliary gradients to the primary task.* Next, we first show the (vanilla) performance of those methods on the basic **Aux-Head** network with a fully shared encoder and separated heads for different tasks, then we report the performance integrating our networks. We conduct semantic segmentation experiments on NYU v2 with surface normal estimation as the auxiliary task.

The results are shown in Table 2, which illustrates that all of the proposed networks without incorporating the optimization-based approaches have already outperformed the best vanilla optimization-based methods. This might imply the superior of the soft parameter sharing architecture methods in dealing with the negative transfer over the optimization-based methods Xin et al. (2022); Shi et al. (2023). Moreover, *our performance can be further improved by incorporating with the optimization-based approaches.*

| NYU v2, Primary: Seg | (%) (↑) | |
|---|---|---|
| | mIoU | PAcc |
| Aux-Head | 34.7 | 65.4 |
| Aux-Head + Uncertainty | 35.2 | 65.6 |
| Aux-Head + DWA | **35.3** | 65.7 |
| Aux-Head + CAGrad | 34.9 | 65.4 |
| Aux-Head + PCGrad | 35.0 | 65.5 |
| Aux-Head + PCGrad-Aux | 35.2 | 65.7 |
| Aux-Head + GCS | **35.3** | **65.9** |
| Aux-G-Stage | 35.4 | 65.9 |
| Aux-G-Layer | 35.6 | 65.9 |
| Aux-NAS | **36.0** | **66.1** |
| Aux-NAS + Uncertainty | 36.0 | 66.3 |
| Aux-NAS + DWA | 36.0 | 66.1 |
| Aux-NAS + CAGrad | 35.9 | 66.0 |
| Aux-NAS + PCGrad | 36.0 | 66.3 |
| Aux-NAS + PCGrad-Aux | 36.2 | 66.3 |
| Aux-NAS + GCS | **36.3** | **66.5** |

Table 2: Comparison with the opt-based methods on NYU v2 semantic seg., with surface normal prediction as auxiliary.

## 4.2 COMPARISON WITH ARCHITECTURE-BASED METHODS

Note that our methods exhibit single task inference cost, which is not ensured in many state-of-the-art architecture-based methods as shown in Table 1. Therefore, besides the **Single** and **Aux-Head** baselines, we compare with **Adashare** Sun et al. (2020) as its inference cost is equal or less than that of a single task. *For more fair comparisons, we implement an auxiliary learning variant of* **Adashare**, where *for the primary task, we fix the select-or-skip policy as all 1's and remove its sparsity regularization. This produces* **Adashare-Aux** *with the same inference cost of ours*.

### 4.2.1 DIFFERENT PRIMARY-AUXILIARY TASK COMBINATIONS

This section is to show that our methods are consistently outperforms the state-of-the-art methods across different primary-auxiliary combinations. We use NYU v2 Silberman et al. (2012) dataset to conduct our experiments. We carry out two different primary-auxiliary semantic segmentation and surface normal estimation combinations, where each of them serves as the primary task and the other is the auxiliary. We use the same network backbone and the same losses as Sect. 4.1. Table 3 show that our methods outperforms the baselines for all the experiments. And our Aux-NAS outperforms Aux-G, as Aux-NAS leverages the additional auxiliary features, which formulates a much larger capacity in the NAS search space, therefore it achieves a better convergence.

### 4.2.2 DIFFERENT DATASETS

To further assess the applicability of our methods across datasets, Table 4 show that our methods consistently outperform other baselines, when using auxiliary monocular disparity estimation to assist the primary semantic segmentation on CityScapes dataset. We also perform object and scene classification tasks on Taskonomy dataset in Appendix D.

### 4.2.3 DIFFERENT BACKBONES

This section is to show the robustness of our methods across single-task backbones. We validates this on both ViTBase with a DPT decoder Ranftl et al. (2021) in Table 5 and ResNet-50 in Appendix E, which demonstrate that the proposed methods consistently outperform other state-of-the-art.

### 4.2.4 SCALABILITY TO MORE AUXILIARY TASKS

*Our method maintains a consistent single-task inference cost with more auxiliary tasks. Moreover, our training complexity scales **linearly** with the number of auxiliary tasks.* This is because i) *linear network Growth*: as we focus solely on the primary task performance, given $K$ auxiliary tasks, we only link $K$ inter-task connections from each auxiliary task to the primary task. This is in contrast to $(K+1)K/2$ inter-task connections for all the task pairs in the soft parameter sharing MTL methods Gao et al. (2019; 2020); Misra et al. (2016); Ruder et al. (2019); ii) *linear NAS complexity w.r.t. the network size*: the single-shot gradient-based NAS algorithm has a linear complexity, as proved in Sect. 2.3 of Liu et al. (2019b). We discuss in details in Appendix A.

We perform surface normal estimation as the primary task on NYU v2 using ViTBase, where we use i) semantic segmentation, and ii) semantic segmentation and depth estimation, as the auxiliary task(s). Table 6 shows that the performance of our method can be further improved with more auxiliary tasks. The average forward-backward time of each 20-sample batch for 1 and 2 auxiliary tasks (*i.e.*, 2 and 3 tasks including the primary task) are 0.337s and 0.556s (roughly 3/2 times of 0.337s), demonstrating the linear scalability of our training complexity to more auxiliary tasks.

| NYU v2, Primary: Normal | Err (↓) | | | Within $t°$ (%) (↑) | |
|---|---|---|---|---|---|
| | Mean | Med. | RMSE | 11.25 | 22.5 |
| Single | 14.6 | 12.9 | 17.7 | 43.2 | 80.8 |
| Aux-Head | 14.8 | 13.2 | 17.9 | 41.9 | 80.1 |
| Adashare | 13.2 | 11.4 | 16.8 | 49.7 | 82.2 |
| Adashare-Aux | 12.9 | 11.0 | 16.7 | 51.9 | 85.5 |
| Aux-G-Layer | 12.6 | 10.7 | 15.7 | 52.3 | **85.9** |
| Aux-NAS | **12.5** | **10.3** | **15.6** | **53.8** | **85.9** |

Table 5: Surface normal prediction on NYU v2 assisted by segmentation using **ViTBase**. We do not have *Aux-G-Stage* as no *Stage* concept in ViTBase.

| NYU v2, Primary: Normal | Err (↓) | | | Within $t°$ (%) (↑) | |
|---|---|---|---|---|---|
| | Mean | Med. | RMSE | 11.25 | 22.5 |
| Aux-G-Layer (1) | 12.6 | 10.7 | 15.7 | 52.3 | 85.9 |
| Aux-NAS (1) | 12.5 | 10.3 | 15.6 | 53.8 | 85.9 |
| Aux-G-Layer (2) | 12.5 | 10.6 | 15.5 | 52.7 | 86.1 |
| Aux-NAS (2) | **12.2** | **10.2** | **15.3** | **54.5** | **86.7** |

Table 6: Scalability to more auxiliary tasks. (1) and (2) denote the number of auxiliary tasks.

## 5 ABLATION ANALYSIS

In this section, we first show the effects of our design, *i.e.* the effects of the Auxiliary Gradients, the Auxiliary Features, with and without the NAS training. We perform ablations using VGG-16 on NYU v2 with semantic segmentation as the primary task, to study the effects of *the auxiliary gradients*, *the auxiliary features*, and the *NAS training*, respectively. Our results in Table 7 demonstrate that i) NAS improves the method which only exploits the auxiliary gradients (*i.e.*, Aux-G-Layer) by discovering the best locations of the primary-to-auxiliary connections, and ii) our Aux-NAS further boost the performance of the Aux-G method with NAS, as it also leverages the auxiliary features.

| Gradient | Feature | NAS | Seg. (%) (↑) | |
|---|---|---|---|---|
| | | | mIoU | PAcc |
| ✓ | | | 35.4 | 65.9 |
| ✓ | | ✓ | 35.7 | 66.0 |
| ✓ | ✓ | ✓ | **36.0** | **66.1** |

Table 7: Effects of the auxiliary gradients, the auxiliary features, and the NAS training.

## 6 CONCLUSION

We aim to exploit the additional auxiliary labels to improve the primary task performance, while keeping the inference as a single task network. We design a novel asymmetric structure which produces changeable architectures for training and inference. We implement two novel methods by exploiting the auxiliary gradients solely, or by leveraging both the auxiliary gradients and features with neural architecture search. Both of our methods converges to an architecture with only primary-to-auxiliary connections, which can be safely removed during the inference to achieve a single-task inference cost for the primary task. Extensive experiments show that our method can be incorporated with the optimization-based approaches, and generalizes to different primary-auxiliary task combinations, different datasets, and different single task backbones.

ACKNOWLEDGMENTS

We thank Haoping Bai for constructive discussion. This work was supported by the National Natural Science Foundation of China (62306214, 62325111, 62372480, 62076099), Natural Science Foundation of Hubei Province of China (2023AFB196), and Knowledge Innovation Program of Wuhan-Shugung Project (2023010201020258).

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

## A DETAILS OF THE TRAINING COST, THE NAS COMPLEXITY, AND THEIR LINEAR SCALABILITY TO MORE AUXILIARY TASKS

We first emphasize that the models for both of our Aux-G and Aux-NAS methods increase linearly with more auxiliary tasks, since we focus solely on the primary task performance. As shown in Fig. 4, given $K$ auxiliary tasks, we only have $K$ inter-task connections from each auxiliary task to the primary task and do not need the interactions between the auxiliary tasks. This is in contrast to $(K+1)K/2$ inter-task connections for all the task pairs in the soft parameter sharing MTL methods Gao et al. (2019; 2020); Misra et al. (2016); Ruder et al. (2019).

**The Auxiliary Learning Architectures (ours):**
It focus solely on the primary task performance and only connects the primary task to each of the auxiliary tasks, resulting in a *linear* inter-task connections w.r.t. the number of auxiliary tasks.

**The Multi-Task Learning Architectures:**
All the input tasks are peers, resulting in a *square* inter-task connections w.r.t. the number of auxiliary tasks, i.e., the inter-task connections are established for all the task pairs.

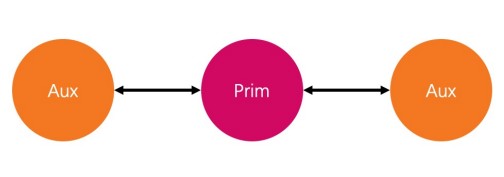
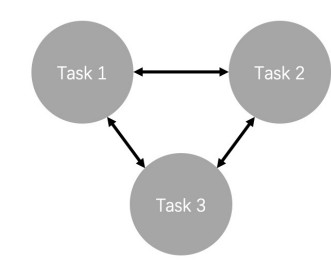

Figure 4: An illustration for the inter-task connections (*i.e.*, the search space) of the auxiliary learning (ours) and the multi-task learning architectures. We use 3 tasks (or 1 primary task plus 2 auxiliary tasks) as an example.

Our Aux-G model is simply trained by gradients, whose complexity is linear w.r.t. the model FLOPs and scales linearly with more auxiliary tasks.

Our Aux-NAS model is trained using the single-shot gradient-based NAS algorithm, as proofs in Sect. 2.3 and Eq. 8 of Liu et al. (2019b), the complexity of the single-shot gradient-based NAS algorithm is linear w.r.t. the amount of the model weights $w$ and that of the architecture weights $\alpha$:

$$O(|w| + |\alpha|). \tag{15}$$

As shown in Fig 4, given $K$ auxiliary tasks, the amount of the model weights becomes $(K+1)|w|$ (including the primary task), and that of the architecture weights becomes $K|\alpha|$ (as we only connect the primary task to each auxiliary task, rather than between the auxiliary task pairs), therefore, the training complexity of our Aux-NAS model is:

$$O((K+1)|w| + K|\alpha|), \tag{16}$$

which scales linear with $K$. Our experiments in Table 6 of the main text also verifies the training time for each batch increases from 0.337s for $K = 1$ to 0.556s for $K = 2$, where the ratio is between (2+1) / (1+1) = 1.5 (for model weights complexity) and 2/1 = 2 (for architecture weights complexity).

## B DETAILS OF THE TASK LOSSES AND THE EVALUATION METRICS

**Losses.** We use the cross-entropy loss for semantic segmentation. Surface normal prediction is trained by the cosine similarity loss. The scale and shift invariant loss Ranftl et al. (2022) is used for depth estimation. We train disparity estimation with the mean squared error (MSE). For the object classification and scene classification on Taskonomy, we use the $\ell_2$ loss as the "groundtruth" given by the Taskonomy dataset is the "soft classification labels" predicted by a large network.

**Evaluation Metrics.** For semantic segmentation, we evaluate with the pixel-wise accuracy (PAcc) and mean intersection over union (mIoU). We use mean, median, and root mean square (RMSE)

| Taskonomy
Primary: Object Cls. | (%) (↑) | |
|---|---|---|
| | Top-1 | Top-5 |
| Single | 34.3 | 65.9 |
| Aux-Head | 34.7 | 66.6 |
| Adashare | 35.9 | 67.1 |
| Adashare-Aux | 36.3 | 67.7 |
| Aux-G-Stage | 37.4 | 67.9 |
| Aux-G-Layer | 37.2 | 68.3 |
| Aux-NAS | **39.8** | **70.7** |

Table 8: Object classification on the Taskonomy dataset with scene classification as the auxiliary task using the **ResNet-50** network.

angle difference, also the percentage of pixels within $11°$ and $22.5°$ w.r.t. the ground truth for surface normal prediction. We report the Top-1 and Top-5 accuracy for object classification.

## C  MORE TRAINING AND EVALUATION DETAILS

**Training Strategy.** We train our **Aux-G** method simply by gradients. For the **Aux-NAS** method, we train $(\alpha^P, \alpha^A)$ and $w$ alternatively by sampling two non-overlapped batches similar to Gao et al. (2020).

**Evaluation Strategy.** We simply cut off all the primary-to-auxiliary connections and the auxiliary branch for inference, which remains a single-task model for the primary task without extra inference computations.

**More Training Details.** We use $321 \times 321$ image samples for the CNN backbones (*i.e.*, ResNet-50 and VGG-16), and $224 \times 224$ image samples for the transformer backbone (*i.e.*, ViTBase Dosovitskiy et al. (2021)). The ViTBase backbone we used is `vit_base_patch16_224` of the huggingface `timm` package. We initialize the single task branches with the pretrained single task model weights. For the fusion operations, we initialize the 1x1 convolution with all 0. We gradually increase $\lambda$ of Eq. 10 from 0 to 100 during the training, and initialize all $\alpha$'s of Eqs. 13 and 14 to 0.5.

For the pixel-labeling tasks (*i.e.*, semantic segmentation, surface normal prediction, depth estimation, and monocular disparity estimation) on CNN backbones (*i.e.*, ResNet-50 and VGG-16), we use a Deeplab head with atrous convolutions Liu et al. (2022). We use a DPT head Ranftl et al. (2021) for those tasks on Transformer backbone (*i.e.*, ViTBase). For the classification tasks (*i.e.*, object classification and scene classification), we simple use a Fully-Connected/MLP layer to map the feature dimensions to the number of classes.

We use the official train/val split for the NYU v2 Eigen & Fergus (2015) and the CityScapes Cordts et al. (2016) datasets. For the Taskonomy Zamir et al. (2018) dataset, we use the official *Tiny* split of it.

We treat the convolutions layers of VGG-16, the residue blocks of ResNet-50, and the multi-head attention blocks of ViTBase as the basic building elements to construct the inter-task connections.

## D  THE EXPERIMENTS ON THE TASKONOMY DATASET

In this section, we perform the object classification task, assisted by the scene classification, on the Taskonomy dataset, using the Res-50 network. The Top-1 and Top-5 recognition rates are reported in Table 8, which, accompanied with Tables 3 and 4 in the main text, demonstrate that our method consistently outperforms the state-of-the-art on various tasks and datasets. 33.8 63.0 37.8 70.5 Multiple 34.1 66.1 37.8 71.2

## E  THE EXPERIMENTS WITH THE RESNET-50 BACKBONE

This section is to further demonstrate the robustness of our methods across backbones. We validates this on a ResNet-50 network in Table 9, which, accompanied with Tables 3 and 5 in the main text,

| NYU v2
Primary: Seg | (%) (↑) | |
|---|---|---|
| | mIoU | PAcc |
| Single | 34.1 | 65.0 |
| Aux-Head | 32.9 | 64.6 |
| Adashare | 33.3 | 64.9 |
| Adashare-Aux | 33.7 | 65.3 |
| Aux-G-Stage | 33.4 | 64.8 |
| Aux-G-Layer | 32.9 | 64.5 |
| Aux-NAS | **36.8** | **66.7** |

Table 9: Semantic segmentation on the NYU v2 dataset with surface normal prediction as the auxiliary task using the **ResNet-50** network.

demonstrate that the proposed methods consistently improve the performance across different single task backbones.

## F ANALYSIS OF THE ARCHITECTURE CONVERGENCE

In the following, we analyze the convergence of the searched architecture based on the Aux-NAS weights of Table 5.

**Auxiliary-to-Primary** Connections: The auxiliary-to-primary architecture weights converges to very small values, with the maximum value is around 0.01, which indicates that our method successfully pruned those **auxiliary-to-primary** down. Notably, all the results reported in our manuscript are the final single-task inference performance, where **we manually set all the auxiliary-to-primary architecture weights to 0 before conducting the evaluation**. This guarantees our objective of ensuring the single-task inference cost.

**Primary-to-Auxiliary** Connections: Regarding the primary-to-auxiliary connections, as we do not impose any regularization on their architecture weights (and those connections will be removed to maintain a single-task inference regardless how they converge), those connections typically converge to soft values between 0 and 1. We conducted 10 replicate runs with random seeds using the same experimental setup as those in Table 5 (*i.e.*, ViTBase on NYUv2), the mean and standard deviation of min, max, mean, median statistics of the primary-to-auxiliary architecture weights are shown below:

| Prim-to-Aux Arch Weight | min | max | mean | median |
|---|---|---|---|---|
| **mean** | 0.1561 | 0.5041 | 0.3073 | 0.2725 |
| **std** | 0.0110 | 0.0120 | 0.0117 | 0.0150 |

Table 10: Statistics of the converged **primary-to-auxiliary** weights. Those results are obtained by performing 10 replicate runs of the experiment in Table 5.

The table above illustrates a small standard deviation of the converged primary-to-auxiliary weights in replicate runs, indicating a degree of stability in our method. We attribute this small deviation in the converged architectures to: i) **for the model weights**, initializing our network backbones with converged single-task networks, and ii) **for the architecture weights**, we focus our architecture search solely on cross-task connections without altering the network backbones.

