# OpenReview forum: "Aux-NAS: Exploiting Auxiliary Labels with Negligibly Extra Inference Cost"
_ICLR.cc/2024/Conference — ICLR 2024 poster_

### Official Review · Reviewer_fJaD · 2023-10-30

**Soundness:** 3 good
**Presentation:** 3 good
**Contribution:** 3 good
**Rating:** 8
**Confidence:** 4

**Summary:**

This paper aims to harness the auxiliary tasks to enhance the performance of the primary task, while maintaining a single task inference cost for the primary task. The authors achieve this by designing an asymmetric network and further develops two algorithms: the first algorithm directly uses the asymmetric primary-to-auxiliary architecture, where the auxiliary tasks can be directly removed during the inference; the second algorithm initiates with an architecture with bi-directional connections, and subsequently exploits a tailored L1 constrained NAS optimization to prune all the auxiliary-to-primary connections, thereby enabling to remove the auxiliary task during inference. The proposed soft-parameter sharing architecture-based method can be integrate with existing optimization-based methods. The author validates their method with extensive experiments on 6 tasks with 3 CNN and transformer architectures.

**Strengths:**

1. This paper formulates the auxiliary learning problem through a task-oriented adaptive feature fusion approach without the need of explicitly identifying the task similarity. Mathematically, such architecture-based method can be seamlessly integrated with a variety of multi-task/auxiliary optimization methods such as loss re-weighting and gradient manipulation. The paper is well written and easy to understand, with an extensive literature review in Table 1 clearly demonstrating the contribution of the proposed method.
2. The evolving and asymmetric network design, coupled with a tailored NAS algorithm, ensures the converged network comprises only the primary-to-auxiliary connections, thereby guaranteeing a single-task inference cost for the learned architecture.
3. Beyond the benefits in the single-task inference cost, the authors also show (in Sect. 4.2.4 and the supplementary) that the training complexity exhibits a linear scalability to multiple auxiliary tasks.
4. The experiments are extensively performed on 6 highly diverse tasks with 3 base net architectures including CNN and transformers. The authors also checked the performance when the primary and the auxiliary tasks possess different architectures in the supplementary. The results of all those experiments are promising.

**Weaknesses:**

1. Is it possible to use Normalization and Activation operations other than BatchNorm and ReLU in Eqs. 13 and 14?
2. In Fig. 3, should the cut-off dash line be between the 1x1 conv and the add operations?
3. I suggest the authors to move the supplementary material into the Appendix of the main text for better readability.

**Questions:**

Please respond to those in the Weakness section.

**Details Of Ethics Concerns:**

I do not have ethics concerns.

---

> ### Author Response · Authors · 2023-11-20
>
> Thank you for your constructive comments. We have thoroughly addressed each of the raised questions with detailed responses. We hope those well resolved your concerns. Please kindly let us know if any further discussion or clarification is needed.
>
> Q1. **Is it possible to use Normalization and Activation operations other than BatchNorm and ReLU in Eqs. 13 and 14?**
>
> It is true that we have the flexibility to leverage different Normalization and Activation operations other than BatchNorm and ReLU. As we are inserting new fusion operations to connect multiple well-trained single-task networks, **the key design principle behind our fusion operations in Eqs. 13 and 14 is to ensure that their insertion does not significantly alter the original well-trained single-task networks**. Therefore, any fusion operations, along with any associated Normalizations and Activations, are applicable as long as they adhere to this fundamental design principle.
>
> Q2. **In Fig. 3, should the cut-off dash line be between the 1x1 conv and the add operations?**
>
> During the NAS optimization, we cut-off the auxiliary-to-primary connections by regularizing $alpha^P$'s in Eq. 13 to 0. Therefore, in Fig. 3, we illustrated the cut-off dash line between the auxiliary input and the concatenation operation. Having said that, it is equivalent to draw the cut-off dash line be between the 1x1 conv and the add operations as suggested, because the output of 1x1 conv is also 0 when the input is 0.
>
> Q3. **I suggest the authors to move the supplementary material into the Appendix of the main text for better readability.**
>
> We have moved the supplementary material, originally in a separate file, to the Appendix of the main text. Additionally, we have included the appropriate references in the main text. Please kindly refer to our updated manuscript.

---

### Official Review · Reviewer_Qzsf · 2023-10-31

**Soundness:** 4 excellent
**Presentation:** 3 good
**Contribution:** 4 excellent
**Rating:** 8
**Confidence:** 4

**Summary:**

This paper introduces a learnable and flexible asymmetric network architecture designed for general-purpose auxiliary learning, where the auxiliary task plays a pivotal role in supporting the primary task's training process, and can be freely removed during the inference. As a result, the proposed method achieves a multi-task level performance while keeping a single-tasks level inference cost. The authors implement their design as adaptive layerwise feature fusion of multiple single-task branches, where the full network converges to an asymmetric architecture with only primary-to-auxiliary connections existed, enabling the removal of the auxiliary task during the inference. Two algorithms are developed to achieve this, where the more advanced one exploits a specifically designed NAS pruning to achieve an asymmetric architecture after convergence. The experiments are extensive across 6 tasks with 3 network backbone architectures, which sufficiently demonstrate the promising performance.

**Strengths:**

1.	This paper tackles the general-purpose auxiliary learning towards a multi-task level performance and a single-tasks level inference cost. The proposed method can be applied to various tasks and network backbones mathematically and also validated experimentally.
2.	The proposed method can also be freely combined with various multi-task or auxiliary task optimization methods listed in Table 1, which was also validated by the experiments.
3.	The single-task level inference cost is assured through the resultant converged asymmetric network architecture. Furthermore, the training cost exhibits a linear increase when incorporating additional auxiliary tasks, which is enabled by the supernet architecture for NAS that only encompasses the connections between the primary task and each of auxiliary tasks.
4.	Table 1 present a very clear and comprehensive taxonomy about the position of the proposed method among the area of multi-task learning and the auxiliary task learning.
5.	The experiments are extensive, validating the generalization on 6 diverse tasks within 3 datasets, and 3 network backbones including both CNNs and Transformers.

**Weaknesses:**

This paper is well written, and I do not see major weakness, but the clarification of the following minor issues would further improve the paper:
1.	I appreciate that the authors provide the full NAS objective in Eq. 10, but the details about how it is optimized need to be further elaborated. If I understand correctly, the model weight w and the architecture weight alpha should be updated iteratively?
2.	I suggest indicating the network backbone in the legends of Tables 3 and 4, as there are several tables in a similar shape that only differ from backbones.
3.	It is suggested to replace the figures with vector images for a better resolution. The paper, in its current version, used a lot of v-spacing; it is also advised to remove them for better readability.

**Questions:**

The author claimed that they implement the tailored version of PCGrad and AdaShare specifically for the auxiliary task learning, i.e., PCGrad-Aux and AdaShare-Aux. What are the details of those auxiliary task learning variants?

---

> ### Author Response · Authors · 2023-11-20
>
> Thank you for your constructive comments. We have thoroughly addressed each of the raised questions with detailed responses. We hope those well resolved your concerns. Please kindly let us know if any further discussion or clarification is needed.
>
> Q1. **Details of NAS optimization.**
>
> It is true that the model weight $w$ and the architecture weight $\alpha$ are updated iteratively during our training of Aux-FG-NAS. Compared to training a fixed neural networks, our NAS optimization requires only one additional forward-backward pass to update the architecture weights in each iteration. This process is straightforward to implement, as indicated in our pseudo code provided above. We will release the complete training and evaluation codes upon acceptance.
>
> Q2. **Indicating the network backbone in the legends of Tables 3 and 4.**
>
> Indeed adding the network backbone into the table legends improves the readability. We followed this suggestion and updated our tables accordingly, please kindly refer to our updated manuscript.
>
> Q3. **Use vector images, remove the v-spacing.**
>
> We thank this suggestion. We have utilized vector images and minimized the v-spacings, please kindly refer to our updated manuscript.

---

### Official Review · Reviewer_V6nv · 2023-11-01

**Soundness:** 3 good
**Presentation:** 3 good
**Contribution:** 3 good
**Rating:** 6
**Confidence:** 3

**Summary:**

The paper introduces a new framework for auxiliary learning, in which the goal is to improve the performance on a task of interest (i.e., primary task) by utilizing auxiliary information. In particular, the proposed method aims to tackle auxiliary learning problems without introducing computational or parameter overhead during inference. To this end, the paper borrows inspiration from multi-task learning and neural architecture search to design an asymmetric network architectures, where the connections from primary-task network layers and auxiliary-task network-layers are directed (from primary to auxiliary), such that computations or parts of networks for auxiliary information can be removed during inference.

**Strengths:**

- The proposed method successfully tackles auxiliary learning without inducing extra computational overhead during inference, by utilizing NAS to design a network that has asymmetric connections directed from primary-task network parts to auxiliary-task network parts.

- The proposed method is flexible in that it can be combined with different auxiliary learning methods

- The paper is clearly written; easy to read and follow.

**Weaknesses:**

- Is there a need to initialize search space to include all bi-directional connections? why not start from networks with only primary-to-auxiliary connections right away?

- Lack of ablation studies related with the question above: the performance change as the search space only contains primary-to-auxiliary connections.

- Missing details: Are all auxiliary-to-primary connections are pruned at the end of training?

- Missing details: What is the final architecture produced by NAS? How consistent is the final performance across different random seeds and trials?

**Questions:**

Written in the weaknesses section.

---

> ### Author Response · Authors · 2023-11-20
>
> Thank you for your constructive comments. We have thoroughly addressed each of the raised questions with detailed responses. We hope those well resolved your concerns. Please kindly let us know if any further discussion or clarification is needed.
>
> Q1.  **The rationale behind using bi-directional NAS initialization. Ablation of the initialization with only primary-to-auxiliary connections.**
>
> **The rationale behind initializing all the bi-directional connections is to exploit both features and gradients from the auxiliary tasks**. During the training, the auxiliary-to-primary connections, which supply auxiliary features, are gradually pruned out during the NAS training to ensure a single-task inference.
>
> We expect that the performance of bi-directional initialization suppresses that of only initializing primary-to-auxiliary connections. This is because **the L1 regularization applied to auxiliary-to-primary connections leads to difficulty in re-establishing them** if we do not initialize them. As a consequence, **the model capacity, i.e., the valid search space, in the primary-to-auxiliary initialization only encompasses a subset of that in the bi-directional initialization**.
>
> To validate the above discussion, we conducted experiments by initializing only primary-to-auxiliary connections as instructed, mirroring the setup in Table 5 using ViTBase on NYUv2. As anticipated, the ablation results align with our expectations, as detailed below.
>
> | NYU v2, Primary: Normal | mean | median | rmse | within 11.25 | within 22.5 | within 30 |
> | ------ | ------ | ------ | ------ | ------ | ------ | ------ |
> | Init. Aux-to-Prim 0.5 (paper) | **12.4553** | **10.3271** | **15.5775** | **53.8358** | **85.9042** | 94.4133
> | Init. Aux-to-Prim 0 | 12.5091 | 10.4472 | 15.6169 | 53.3136 | 85.6822 | **94.4466** |
>
> Q2. **Ablation of the search space that only contains primary-to-auxiliary connections.**
>
> In addition to the ablation in the previous question where **the search space contains bi-directional connections but auxiliary-to-primary links were initialized to 0**, we also perform another ablation following your guidance, where **the search space only contains primary-to-auxiliary connections**. The results are listed below, which exhibit inferior performance w.r.t. our original Aux-FG-NAS. Also notably, the single-side search space setting here performs very much similarly to Q1 where auxiliary-to-primary links were initialized to 0. This outcome comes from similar reasons as discussed in the previous question, i.e., **the model capacity, and the search space, of the single-directional setting are just a subset of our original bi-directional Aux-FG-NAS**.
>
> | NYU v2, Primary: Normal | mean | median | rmse | within 11.25 | within 22.5 | within 30 |
> | ------ | ------ | ------ | ------ | ------ | ------ | ------ |
> | Bi-directional Search Space (paper) | **12.4553** | **10.3271** | **15.5775** | **53.8358** | **85.9042** | **94.4133**
> | Single-directional Prim-to-Aux Search Space | 12.5443 | 10.4538 | 15.6595 | 53.3297 | 85.6684 | 94.2703 |
>
> Q3.  **Are all auxiliary-to-primary connections pruned at the end of training?**
>
> We have collected the statistics of the auxiliary-to-primary architecture weights for all of our experiments, revealing that **the maximum value of all the auxiliary-to-primary architecture weights is 0 at the precision of 10^-2**.
>
> Notably, all the results reported in our manuscript are the final single-task inference performance, where **we manually set all the auxiliary-to-primary architecture weights to 0 before conducting the evaluation**. This guarantees our objective of ensuring the single-task inference cost.

---

> ### Author Response · Authors · 2023-11-20
>
> Q4.  **What is the final architecture produced by NAS? How consistent is the final performance across different random seeds and trials?**
>
> It is indeed interesting to see the final convergence of the learned architecture.
>
> As discussed in the previous question, the auxiliary-to-primary connections are pruned out with the maximum architecture weights less than 0.01.
>
> Regarding the primary-to-auxiliary connections, as we do not impose any regularization on their architecture weights (and those connections will be removed to maintain a single-task inference regardless how they converge), **those connections typically converge to soft values between 0 and 1**. Therefore, it is difficult to draw them clearly for illustration, given the amount of those connections and many replicate runs. Consequently, we instead gather their statistics of min, max, mean, median of 10 replicate runs with random seeds, and show the standard deviation of those statistics.
>
> Specifically, we conducted the same experiments as those in Table 5 using ViTBase on NYUv2, with random seeds for 10 replicate runs. The mean and standard deviation of min, max, mean, median statistics of the primary-to-auxiliary architecture weights are shown below:
>
> | NYU v2, Primary: Normal, Prim-to-Aux Arch Weight | min | max | mean | median |
> | ------ | ------ | ------ | ------ | ------ |
> | mean | 0.1561 | 0.5041 | 0.3073 | 0.2725 |
> | std | 0.0110 | 0.0120 | 0.0117 | 0.0150 |
>
> The above table demonstrates negligible standard deviation, which further produces subtle differences in the final performances, as shown in Question 3 of Reviewer CQEV. We believe that the subtle standard deviation in both the converged architectures and the final performances is ensured by: i) **for the model weights**, our network backbones are initialized by the converged single-task networks, and ii) **for the architectures**, we only search the cross-task connections, rather than altering the network backbones.
>
> We include the raw primary-to-auxiliary architecture statistics in the following thread. We note that all the results presented here in the rebuttal and in our main text are obtained without specifying any seeds. We will release our training codes also without the specification of seeds.
>
> We thank this discussion again, and please kindly let us know if any further discussion or clarification is needed.

---

> ### Author Response · Authors · 2023-11-20
> **Raw data for Question 4:**
>
> The raw data of converged **primary-to-auxiliary** architecture weights for Question 4. **The maximum value of all the auxiliary-to-primary architecture weights is 0 at the precision of 10^-2**.
> | NYU v2, Primary: Normal, Prim-to-Aux Arch Weight | min | max | mean | median |
> | ------ | ------ | ------ | ------ | ------ |
> | Trial 1 | 0.161 | 0.534 | 0.299 | 0.275 |
> | Trial 2 | 0.142 | 0.495 | 0.296 | 0.263 |
> | Trial 3 | 0.136 | 0.492 | 0.294 | 0.256 |
> | Trial 4 | 0.151 | 0.503 | 0.303 | 0.261 |
> | Trial 5 | 0.152 | 0.497 | 0.301 | 0.266 |
> | Trial 6 | 0.170 | 0.511 | 0.310 | 0.267 |
> | Trial 7 | 0.158 | 0.504 | 0.317 | 0.284 |
> | Trial 8 | 0.165 | 0.506 | 0.304 | 0.263 |
> | Trial 9 | 0.158 | 0.504 | 0.317 | 0.284 |
> | Trial 10 | 0.168 | 0.495 | 0.332 | 0.306 |

---

> > ### Comment · Reviewer_V6nv · 2023-11-21
> >
> > Thank you for the rebuttal.
> > While most of my concerns are addressed, the weight values of aux-to-prim connections are quite high, considering the usual values of weights. So, I'm quite confused as to how the performance is still preserved after removing connections. How are the weight values of prim-to-prim connections? Relative comparisons will give better picture of what's happening. Also, how much performance drop occurs after removing aux-to-prim connections?

---

> ### Author Response · Authors · 2023-11-21
>
> Dear Reviewer,
>
> We sincerely apologize for the typos in our initial table entry for Q4 and its corresponding raw data table. The correct term should be **Prim-to-Aux Arch Weight** instead of **Aux-to-Prim Arch Weight**. We have rectified these errors. Our original text response of Q4 does not have those typos.
>
> We gathered the maximum value of all the **Aux-to-Prim** architecture weights, **which is 0 at the precision of 10^-2**, please also refer to a more detailed discussion about the **Aux-to-Prim** architecture weights in Q3.
>
> We re-evaluated our Aux-FG-NAS model of Table 5, without manually removing the **Aux-to-Prim** connections, the differences are subtle as shown below:
> | NYU v2, Primary: Normal | mean | median | rmse | within 11.25 | within 22.5 | within 30 |
> | ------ | ------ | ------ | ------ | ------ | ------ | ------ |
> | w/ Manual removal of **Aux-to-Prim**  (paper) | 12.4553 | 10.3271 | 15.5775 | 53.8358 | **85.9042** | 94.4133
> | w/o Manual removal of **Aux-to-Prim** | **12.4253** | **10.3101** | **15.5594** | **53.8829** | 85.8975 | **94.4482** |
>
> We apologize again for the confusion and appreciate the opportunity for further discussion. Please kindly let us know if any additional clarification is needed.

---

> ### Comment · Reviewer_V6nv · 2023-11-21
>
> Thanks for the clarification.
> My concerns have been addressed if the discussions are included in the revised paper.
> Also, please compare the weights of aux-to-prim connections with those of prim-to-prim to show better contrast.

---

> > ### Author Response · Authors · 2023-11-21
> >
> > We appreciate your suggestion, we will include the following results in our updated Appendix in a few hours:
> > 1. Statistics of both converged **aux-to-prim** and **prim-to-aux** architecture weights.
> > 2. Performance comparison with and without manually removing the **aux-to-prim** connections.
> >
> > Thank you again for your valuable comments!

---

> > > ### Comment · Reviewer_V6nv · 2023-11-21
> > >
> > > Could you please include the weights of prim-to-prim connections as well as mentioned in my response above? This may help for better understanding of relative importance of weights of networks that remain and weights that are not used during inference.

---

> > > > ### Author Response · Authors · 2023-11-21
> > > >
> > > > Thank you for your additional comments. We would be happy to do that but we are less certain whether the **prim-to-prim** connection is a typo?
> > > >
> > > > Our Aux-FG-NAS method includes only two types of cross-task connections, where:
> > > >
> > > > 1. **prim-to-aux** connections: which are used to exploit the auxiliary gradients to train the primary network. They can be freely removed during the inference of the primary task. Therefore, **they are retained during the training** without imposing any regularization during NAS optimization. The architecture weights for these connections converge to soft values between 0 and 1, as we discussed in Q4 and the corresponding raw data table.
> > > >
> > > > 2. **aux-to-prim** connections: which are used to leverage the auxiliary features during the primary network training. **They are gradually pruned out** by L1 regularization in NAS, ensuring the converged architecture attains a single task inference cost for the primary task. The architecture weights for these connections converge to 0 at the precision of 10^-2, as we discussed in Q3. Manually removing those connections only yields subtle differences in results, as we discussed in the above thread today.
> > > >
> > > > We would like to include i) our discussion in Q3 and Q4, ii) the statistics of both converged **prim-to-aux** and **aux-to-prim** weights, and iii) the verification of the subtle performance drop (today's above table) in our updated appendix. Those are able to well analyze "*the relative importance of weights of networks that remain and weights that are not used during inference*".
> > > >
> > > > We hope this clarification enhances the understanding of our discussion.

---

> > > > > ### Comment · Reviewer_V6nv · 2023-11-21
> > > > >
> > > > > By prim-to-prim, I meant weights of primary-task network, which are the ones that contribute to the final performance at test time. I just thought it would be interesting to see the ratio of the weight that is being used for inference (i.e., weight of primary-task network) and those being pruned (aux-to-prim connections).

---

> > > > > > ### Author Response · Authors · 2023-11-21
> > > > > >
> > > > > > Thank you for the clarification. Yes, we can gather the statistics of the weights of the primary-task network, as shown below (collected from the Aux-FG-NAS model in Table 5):
> > > > > >
> > > > > > | NYU v2, Primary: Normal, Primary task model weights | min | max | mean | median |
> > > > > > | ------ | ------ | ------ | ------ | ------ |
> > > > > > | weights | -1.6339 | 18.7777 | -1.2572 | 0.0003 |
> > > > > > | biases | -25.3467 | 11.8667 | -0.1079 | -0.6597 |
> > > > > >
> > > > > > We would like to attach a reminder that the architecture weight is **either 0 or 1** in the seminal discrete NAS (e.g., RL-based). While in the gradient-based NAS, the relaxed continuous architecture weight represents the contribution of a network operation within the search space to the discovered network, **which also typically falls in [0, 1]**. On the other hand, the model weights, including conv/attention weights and biases, **do not adhere to such a constrained range**. Therefore, we feel that a direct comparison between architecture weights and model weights may not be appropriate.
> > > > > >
> > > > > > We hope this further addresses your concern, we are happy to discuss more if any further clarification is needed.

---

> ### Author Response · Authors · 2023-11-21
> **Follow-ups of the “prim-to-prim connections” issue**
>
> Following-up with our previous response, we would like to clarify that **our search space only includes the cross-task *prim-to-aux* and *aux-to-prim* connections**, we leave the architectures of the single-task backbones intact (equivalent to fix the **prim-to-prim** and **aux-to-aux** architecture weights to 1). This ensures the adaptability of our methods to diverse single-task backbones.
>
> As acknowledge by Reviewer CQEV, our Aux-FG-NAS can be further extended to search for a better backbone, which can optionally exploits the priors of specific input single-task backbones and/or the characteristic of input tasks. But this falls outside the scope of our current focus, which is to design **a general-purpose auxiliary learning method that adapts to diverse tasks, single-task backbones, datasets, and integrates with various Multi-Task Optimization methods**.
>
> We hope this well addresses your concerns, please let us know if further discussion is needed and we are happy to engage into that.

---

> > ### Comment · Reviewer_V6nv · 2023-11-23
> >
> > Thanks for the clarification. My questions and concerns have been addressed.

---

### Official Review · Reviewer_CQEV · 2023-11-01

**Soundness:** 3 good
**Presentation:** 3 good
**Contribution:** 3 good
**Rating:** 8
**Confidence:** 4

**Summary:**

This paper presents an architecture based take on auxiliary learning. They recognize that the asymmetry between the auxiliary and primary tasks can be exploited by learning architectures with constraints that favor transfer of information from the auxiliary to the primary, but in an indirect way so as to minimize the possibility / effect of negative transfer.

**Strengths:**

1. The idea is novel and interesting.  I think the use of joint training, followed by the slow trimming of the aux-to-prim connections via L1 regularization is a clever way of more intimately introducing the auxiliary task  but remove it later to avoid needing it during inference.
2. The paper is clearly written and easy to follow
3. This has interesting implications for Auxiliary learning based architecture search -- since what was searched for in this paper were connections, there are expansions on this that can focus on other parts of the architecture space.

**Weaknesses:**

1. Method might be a bit too complex / cumbersome to be practically implemented widely -- especially given the size of the gains.
2. Method also significantly increases memory / compute overhead at training time
3. The experimental results have no error-bars. It's thus hard to judge the significance of the results

### Nitpicks
1. The introduction has *a lot* of italicized text, many of which I think are unnecessary and distracting.

### Some relevant papers
On gradient conflict
1. Dery, Lucio M., Yann Dauphin, and David Grangier. "Auxiliary task update decomposition: The good, the bad and the neutral." arXiv preprint arXiv:2108.11346 (2021).
2. Royer, Amelie, Tijmen Blankevoort, and Babak Ehteshami Bejnordi. "Scalarization for Multi-Task and Multi-Domain Learning at Scale." arXiv preprint arXiv:2310.08910 (2023).

On NAS-like construction of auxiliary objectives
1. Dery, Lucio M., et al. "AANG: Automating Auxiliary Learning." arXiv preprint arXiv:2205.14082 (2022).

**Questions:**

1. Did you try further finetuning the final model on the primary task only after being done with the auxiliary-task based NAS ? This could result in extra performance boost

---

> ### Author Response · Authors · 2023-11-20
>
> Thank you for your constructive comments. We have thoroughly addressed each of the raised questions with detailed responses. We hope those well resolved your concerns. Please kindly let us know if any further discussion or clarification is needed.
>
> Q1.  **Method might be a bit too complex/cumbersome to be practically implemented widely -- especially given the size of the gains.**
>
> We proposed two general-purpose auxiliary learning methods. The first one, Aux-G, is very simple, involving only the direct integration of the primary and auxiliary networks through primary-to-auxiliary connections. The combined network weights can then be trained directly using gradients, akin to training most neural networks with fixed architecture.
>
> The second method, Aux-FG-NAS, leverages Neural Architecture Search for further improved performance. Notably, the chosen NAS optimization method, DARTS, is also easy to implement, **requiring only one additional forward-backward pass in each iteration to train the architecture**. As shown in our pseudo codes provided above, such an additional step can be implemented with **just 7 lines of codes in PyTorch**. We will release the complete training and evaluation codes upon acceptance.
>
> Regarding the assessment of gains which can be subjective, it is important to acknowledge the challenge of designing a general-purpose auxiliary learning framework that seamlessly adapts to diverse tasks, network backbones, datasets, and integrates with various Multi-Task Optimization methods. Notably, in many of our experiments, the improvement achieved by our method versus the previous SOTA surpasses that of the previous SOTA versus the simple Aux-Head baseline. We also appreciate your acknowledgment in Strength 3 regarding the potential of our method to further exploit the task prior in network backbones for additional enhancements.
>
> Q2.  **Method also significantly increases memory / compute overhead at training time.**
>
> **The training memory / compute complexity for both of our methods scales to multiple auxiliary tasks linearly**, which aligns with the Soft-Parameter Sharing Multi-Task Architectures (SPS-MTA) and most of Multi-Task Optimizations (MTO) that employ gradient manipulation (Note that MTO with gradient manipulation has linear instead of constant scalability because their training procedure requires to retain K copies of backward graph for storing and manipulating the gradients from K tasks).
>
> Specifically, given K auxiliary tasks, denote model weights as $w$ and architecture weights as $\alpha$, **our Aux-G method has the same training complexity as that of SPS-MTA methods and most MTO methods**.
>
> **While for our Aux-FG-NAS, the training complexity is** $O((K+1)|w| + K|\alpha|)$. This linear scalability is achieved by the efficient NAS optimization of DARTs [1], as detailed in Eq. 8 of [1], and is confirmed by our pseudo codes provided in the above response.
>
> Further details of our Aux-FG-NAS complexity can be found in Sect. 4.2.4 and Appendix A of our updated manuscript (Appendix A was originally in a separate supplementary file at the time of the initial submission).
>
> It's noteworthy that our method is primarily designed to address practical scenarios where **the inference complexity takes precedence over the training complexity**. Such scenarios are prevalent in customer-facing web applications where the model is only trained for a few times, then deployed online and subjected to million to billion inferences. While for the scenarios where the training resource is also constrained, we can at least use the Aux-G method with the same training complexity as SPS-MTA and most MTO methods, so as to ensure the improvement leveraging auxiliary tasks.
>
> Q3.  **Error bar in the results.**
>
> Following your guidance, we perform 10 replicate runs with random seeds of the ViTBase experiments on NYU v2, as those in Table 5 of our manuscript. We did not observe a substantial standard deviation in the results:
>
> | NYU v2, Primary: Normal | mean | median | rmse | within 11.25 | within 22.5 | within 30 |
> | ------ | ------ | ------ | ------ | ------ | ------ | ------ |
> | mean | 12.5656 | 10.4620 | 15.6895 | 53.3011 | 85.6956 | 94.2751 |
> | std | 0.0479 | 0.0323 | 0.0702 | 0.1341 | 0.2018 | 0.1475 |
>
> We believe that the subtle standard deviation in the replicate runs is ensured by: i) **for the model weights**, our network backbones are initialized by the converged single-task networks, and ii) **for the architectures**, we only search the cross-task connections, without altering the network backbones.
> Notably, all the results presented in our manuscripts and here in the rebuttal are obtained with random seeds. Additionally, our training codes will also be released without any specification of seeds.
> We put the raw results of the 10 replicate runs in the following thread.
> We thank this suggestion again and hope this addressed your concern.

---

> ### Author Response · Authors · 2023-11-20
>
> Q4.  **The unnecessary italicized text in the introduction.**
>
> We thank for pointing this out. We have removed unnecessary italicized text, please see our updated manuscript.
>
> Q5.  **Relevant papers [4,5,6].**
>
> We appreciate for guiding us to those relevant papers, this certainly makes our taxonomy in Table 1 more comprehensive and self-contained. We have cited [4,6] in MTL & AL Methods - Optimization-based - For AL, and [5] in MTL & AL Methods - Optimization-based - For MTL, in Table 1. We also cite and discuss them in our related work section. Thanks again for pointing them out!
>
> Q6.  **Further finetune the final model on the primary task after NAS converges.**
>
> As instructed, we continue to finetune the Aux-FG-NAS weights of Table 5 (i.e., the ViTBase NYUv2 experiment). We use the same learning rate and the same optimizer as those used to further train the converged single-task initializations. The results are presented below, showing a slight (but not significant) improvement over the original results. We thank this advice again.
>
> | NYU v2, Primary: Normal | mean | median | rmse | within 11.25 | within 22.5 | within 30 |
> | ------ | ------ | ------ | ------ | ------ | ------ | ------ |
> | Org results (paper) | 12.4553 | 10.3271 | 15.5775 | 53.8358 | **85.9042** | 94.4133
> | Further finetuned results | **12.3787** | **10.3074** | **15.4840** | **53.9111** | 85.8241 | **94.5066** |
>
>
> **References**
>
> [1] Hanxiao Liu, Karen Simonyan, and Yiming Yang. DARTS: Differentiable architecture search. In ICLR, 2019.
>
> [2] Sirui Xie, Hehui Zheng, Chunxiao Liu, Liang Lin. SNAS: Stochastic Neural Architecture Search. In ICLR, 2019.
>
> [3] Guohao Li, Guocheng Qian, Itzel C. Delgadillo, Matthias Müller, Ali Thabet, Bernard Ghanem. SGAS: Sequential Greedy Architecture Search. In CVPR, 2020.
>
> [4] Lucio M. Dery, Yann Dauphin, David Grangier. Auxiliary task update decomposition: the good, the bad and the neutral. In ICLR, 2021.
>
> [5] Amelie Royer, Tijmen Blankevoort, Babak Ehteshami Bejnordi. Scalarization for Multi-Task and Multi-Domain Learning at Scale. In NeurIPS, 2023.
>
> [6] Lucio M. Dery, Paul Michel, Mikhail Khodak, Graham Neubig, Ameet Talwalkar. AANG: Automating Auxiliary Learning. In ICLR, 2023.

---

> > ### Author Response · Authors · 2023-11-20
> > **Raw data for Question 3:**
> >
> > | NYU v2, Primary: Normal | mean | median | rmse | within 11.25 | within 22.5 | within 30 |
> > | ------ | ------ | ------ | ------ | ------ | ------ | ------ |
> > | Trial 1 | 12.6284 | 10.4828 | 15.7873 | 53.1855 | 85.4142 | 94.0553 |
> > | Trial 2 | 12.5728 | 10.4328 | 15.7136 | 53.4212 | 85.5225 | 94.1904 |
> > | Trial 3 | 12.5767 | 10.4205 | 15.7248 | 53.4658 | 85.6107 | 94.1909 |
> > | Trial 4 | 12.5360 | 10.5016 | 15.6206 | 53.1639 | 85.9002 | 94.4342 |
> > | Trial 5 | 12.5597 | 10.4921 | 15.6675 | 53.1784 | 85.7603 | 94.3500 |
> > | Trial 6 | 12.5745 | 10.4546 | 15.7024 | 53.3280 | 85.5735 | 94.2196 |
> > | Trial 7 | 12.5049 | 10.4366 | 15.6060 | 53.4114 | 85.9511 | 94.4676 |
> > | Trial 8 | 12.6538 | 10.5119 | 15.8035 | 53.0889 | 85.4525 | 94.1168 |
> > | Trial 9 | 12.5049 | 10.4366 | 15.6060 | 53.4114 | 85.9511 | 94.4676 |
> > | Trial 10 | 12.5440 | 10.4506 | 15.6634 | 53.3570 | 85.7202 | 94.2592 |

---

> > ### Comment · Reviewer_CQEV · 2023-11-21
> > **Thanks for the detailed response !**
> >
> > Thanks for responding to all my questions !
> > I have no further clarification questions

---

### Official Review · Reviewer_fF8U · 2023-11-02

**Soundness:** 2 fair
**Presentation:** 3 good
**Contribution:** 2 fair
**Rating:** 6
**Confidence:** 3

**Summary:**

The paper studies how to harness additional auxiliary labels from an auxiliary task to elevate the performance of the main task without escalating the inference cost. To do so, the authors propose to employ individual networks for different tasks and only regularize the main task with the auxiliary task’s gradient. It’s understandable that this act allows the network trained on the auxiliary task to be completely pruned during inference. Furthermore, the authors propose to search for the most appropriate structure that satisfies the previously mentioned constraint with NAS. The paper accentuates its methodology's compatibility with prevailing optimization-based auxiliary learning techniques. The empirical validation, evident from experiments on NYU v2, CityScapes, and Taskonomy datasets using well-known backbones like VGG-16, ResNet-50, and ViT-B, demonstrates the efficacy of the proposed method.

**Strengths:**

++ The paper is well-written with clearly motivated arguments and insights. The auxiliary learning task is also meaningful when we only seek to boost one task with another and aim at quick inference.

++ Table 1 provides a comprehensive understanding and meticulous survey of the field. The authors offer an exhaustive overview of both Multi-Task Learning (MTL) and Auxiliary Learning (AL) methods. Authors have incorporated a wide range of references from multiple years, indicating a holistic survey of both seminal works and recent advancements. The inclusion of their method alongside existing techniques also provides clarity on its positioning within the broader research landscape.

++ The proposed method is backbone- and task-agnostic that is applicable to multiple backbones and tasks.

**Weaknesses:**

-- I am not very familiar with auxiliary learning. However, I do think one baseline might be meaningful, which is to share a single backbone while projecting the gradient of the auxiliary task to the orthogonal direction of the main task on all (or selective) layers. This baseline also has no inference lag while exploiting the auxiliary objective signals.

-- The authors use NAS to search for suitable architectures to optimize the main task's objective. However, distinct backbones are used for each task, and their weights can vary significantly. I'm uncertain why "stitching" two backbones with different weights and objectives is logical. Are there any supporting theories or references?

-- The authors claim that their method achieves "promising performance." However, based on Tables 3 and 4, it appears that the performance gain of the proposed method is only marginal. Considering the additional training costs in the NAS search and optimization, I am not sure whether the loss in training efficiency is worth it.

**Questions:**

See weakness.

---

> ### Author Response · Authors · 2023-11-20
>
> Thank you for your constructive comments. We have thoroughly addressed each of the raised questions with detailed responses. We hope those well resolved your concerns. Please kindly let us know if any further discussion or clarification is needed.
>
> Q1.  **The baseline of sharing a single backbone, while projecting the gradient of the auxiliary task according to the main task on all (or selective) layers.**
>
> The suggested method is indeed a stronger baseline by using Multi-Task Optimization techniques. In fact, the suggested method is exactly what we refer to as **PCGrad-Aux** in Table 2. Specifically, we extend the original PCGrad [1] to auxiliary learning in PCGrad-Aux exactly by the way as suggested, i.e., fixing the primary gradient and projecting all auxiliary gradients w.r.t. the primary gradient, instead of randomly choosing a task for projection in the original PCGrad [1].
>
> Moreover, we also validate our method using another Multi-Task Optimization method specifically designed for auxiliary learning named **GCS** [2]. In GCS, the auxiliary gradients are applied only if they exhibit a non-negative cosine similarity with the primary gradients; otherwise, GCS discards the auxiliary gradients.
>
> The experimental results w.r.t. the auxiliary learning-based Multi-Task Optimization PCGrad-Aux and GCS from Table 2 are:
>
> | NYU v2, Primary: Seg | Aux-Head | Aux-Head + PCGrad-Aux | Aux-Head + GCS | Aux-FG-NAS | Aux-FG-NAS + PCGrad-Aux | Aux-FG-NAS + GCS |
> | ------ | ------ | ------ | ------ | ------ | ------ | ------ |
> | mIoU | 34.7 | 35.2 | 35.3 | 36.0 | 36.2 | 36.3 |
> | PAcc | 65.4 | 65.7 | 65.9 | 66.1 | 66.3 | 66.5 |
>
> Our results, as presented above and in Table 2, demonstrate that i）**Our vanilla method surpasses those Multi-Task Optimization methods**, i.e., Aux-FG-NAS vs. Aux-Head + PCGrad-Aux & Aux-Head + GCS. ii) Importantly, **our method is seamlessly incorporable with these Multi-Task Optimization methods**, yielding further improvement in results, as demonstrated by Aux-FG-NAS vs. Aux-FG-NAS + PCGrad-Aux & Aux-FG-NAS + GCS.
>
> We note that the second point was also acknowledged by Reviewers V6nv, Qzsf, fJaD. Therefore, we focus on comparing our method with other Multi-Task Architecture methods in the subsequent experiments to avoid potential distractions.
> We thank this discussion again and hope this resolved your concern.
>
> Q2.  **Why "stitching" two backbones with different weights and objectives is logical? Are there any supporting theories or references?**
>
> Our method stitches the primary and the auxiliary networks with different weights and objectives, where **the auxiliary networks can be regarded as a "*loss function*" to provide gradients, as well as a feature generator to offer features from another view** (though those auxiliary features are pruned out after convergence). Moreover, during the training, **we refrain from imposing any regularization that enforces similarity between the weights of the primary and the auxiliary networks**. This deliberate choice allows us to stitch the primary and auxiliary networks with different weights and objectives, enabling the auxiliary networks to contribute unique "*losses*" and "*features*" distinct from the primary network.
>
> We also note that the practice of stitching two backbones with different weights and objectives has a longstanding history in architecture-based MTL [3,4,5,6], where one of the seminal works happened to be termed as *cross-stitching networks* [3].
>
> Q3.  **The marginal improvement of Table 3 and Table 4 given the loss in training efficiency.**
>
> While the assessment of the improvement can be subjective, we would like to highlight that **our primary goal is to establish a general-purpose auxiliary-learning framework**. This framework is designed to be seamlessly incorporable with diverse tasks, network backbones, datasets, and various Multi-Task Optimization methods, in a plug-and-play manner. In pursuit of this generality, **we specifically concentrate on the cross-task connections while leaving the network backbone intact**. As acknowledged by Reviewer CQEV (Strength 3), **we have the potential to leverage the task priors, optionally by searching the task backbones, to further enhance the performance**.
>
> Moreover, it is worth noting that in Tables 3 and 4, the improvement of the previous SOTA over the simple Aux-Head baseline is notably less significant than that of our method over the previous SOTA. This observation also underscores the inherent challenges in designing a general-purpose auxiliary learning framework.
>
> As for the training efficiency, please also kindly refer to the above pseudo codes and Question 2 of Reviewer CQEV.

---

> > ### Author Response · Authors · 2023-11-20
> >
> > **References**
> >
> > [1] Tianhe Yu, Saurabh Kumar, Abhishek Gupta, Sergey Levine, Karol Hausman, and Chelsea Finn. Gradient surgery for multi-task learning. In NeurIPS, 2020.
> >
> > [2] Yunshu Du, Wojciech M Czarnecki, Siddhant M Jayakumar, Razvan Pascanu, and Balaji Lakshminarayanan. Adapting auxiliary losses using gradient similarity. arXiv preprint arXiv:1812.02224v2, 2020.
> >
> > [3] Ishan Misra, Abhinav Shrivastava, Abhinav Gupta, and Martial Hebert. Cross-stitch networks for multi-task learning. In CVPR, 2016.
> >
> > [4] Yuan Gao, Jiayi Ma, Mingbo Zhao, Wei Liu, and Alan L Yuille. NDDR-CNN: Layerwise feature fusing in multi-task cnns by neural discriminative dimensionality reduction. In CVPR, 2019.
> >
> > [5] Sebastian Ruder, Joachim Bingel, Isabelle Augenstein, and Anders Søgaard. Latent multi-task architecture learning. In AAAI, 2019.
> >
> > [6] Yuan Gao, Haoping Bai, Zequn Jie, Jiayi Ma, Kui Jia, and Wei Liu. MTL-NAS: Taskagnostic neural architecture search towards general-purpose multi-task learning. In CVPR, 2020.

---

### Author Response · Authors · 2023-11-20
**Response to ACs and all the reviewers**

We sincerely appreciate the AC and the reviewers for their great efforts and time in providing the valuable comments.

In addition to addressing each specific question comprehensively, we notice that Reviewers fF8U, CQEV, and Qzsf concerned about our training efficiency, the complexity of our method, or the NAS optimization procedure. Therefore, we would like to provide the pseudo codes for our advanced Aux-FG-NAS method. We will release the full training and evaluation codes upon acceptance.
```sh
# Model weights training, exactly the same as those when we train a network normally:
X, Y_normals, Y_segmentations = next(iter_train_network_data)       # sample a batch
output_normals, output_segmentations = self.model(X)       # forward pass
normal_loss, seg_loss = self.get_loss_normal_seg(output_normals, output_segmentations, Y_normals, Y_segmentations)       # task losses
loss = normal_loss * self.config['normal_loss_weight'] + seg_loss * self.config['seg_loss_weight']
loss.backward()       # backward pass
self.optimizer_network.step()       # update model weights, self.optimizer_network stores the model weights.

# Arch weights training, one simple extra forward-backward pass to train the arch weights:
X, Y_normals, Y_segmentations = next(iter_train_arch_data)       # sample another non-overlap batch
output_normals, output_segmentations = self.model(X)       # forward pass
normal_loss, seg_loss = self.get_loss_normal_seg(output_normals, output_segmentations, Y_normals, Y_segmentations)       # task losses
l1_loss = self.model.regu_loss()       # L1 loss to prune the auxiliary-to-primary links
loss = normal_loss * self.config['normal_loss_weight'] + seg_loss * self.config['seg_loss_weight'] + l1_loss * self.config['l1_loss_weight']
loss.backward()       # backward pass
self.optimizer_arch.step()       # update arch weights, self.optimizer_arch stores the arch weights.
```
In summary:
* Our advanced Aux-FG-NAS is easy to implement, which **requires only one additional forward-backward pass in each iteration to train the architecture**. This extra step can be conveniently accomplished with **just 7 lines of codes in PyTorch**, as illustrated in the second half of the provided pseudo codes. And as shown in the pseudo codes, we train the model weights and the architecture weights iteratively in each iteration.
* With $K$ auxiliary tasks, model weights $w$, and architecture weights $\alpha$, our Aux-FG-NAS method exhibit a training complexity of $O((K+1)|w| + K|\alpha|)$. Notably, **our Aux-FG-NAS scales linearly with multiple auxiliary tasks**, akin to existing Soft-Parameter Sharing Multi-Task Architecture (SPS-MTA) methods and most Multi-Task Optimization (MTO) methods with gradient manipulation. Please refer to Appendix A of our updated manuscript for details (where those content was originally placed in the separated supplementary file at the time of initial submission).
* Our Aux-G method is very simple and can be trained using the first half of the pseudo codes, akin to most neural networks with fixed architecture. Importantly, **Aux-G shares the same training complexity as that of SPS-MTA and most MTO methods**.

---

### Meta-Review · Area_Chair_gfcg · 2023-12-05

**Metareview:**

The final ratings for this work are: marginally above the acceptance threshold $\times$ 2, accept $\times$ 3. It is evident that all reviewers hold a favorable attitude towards accepting this work, with three reviewers explicitly indicating acceptance. After a thorough review of the reviewers' comments, most of them acknowledge the applicability and effectiveness of the proposed method. Additionally, during the response phase, the authors provided sufficiently detailed explanations addressing the concerns raised by the reviewers. The responses from the authors effectively alleviated the concerns raised by the reviewers. Overall, considering the positive feedback and performance in multiple aspects, I decide to accept this work.

**Justification For Why Not Higher Score:**

Despite the positive stance from the reviewers towards this work, a series of comments in the discussion phase suggests that there might be room for improvement in detailing the theoretical aspects of the work. Therefore, I have chosen not to further elevate my final rating.

**Justification For Why Not Lower Score:**

Please refer to the metareview.

---

### Decision · Program_Chairs · 2024-01-16

Accept (poster)